# ErbB3 Phosphorylation as Central Event in Adaptive Resistance to Targeted Therapy in Metastatic Melanoma: Early Detection in CTCs during Therapy and Insights into Regulation by Autocrine Neuregulin

**DOI:** 10.3390/cancers11101425

**Published:** 2019-09-25

**Authors:** Ciro Francesco Ruggiero, Debora Malpicci, Luigi Fattore, Gabriele Madonna, Vito Vanella, Domenico Mallardo, Domenico Liguoro, Valentina Salvati, Mariaelena Capone, Barbara Bedogni, Paolo Antonio Ascierto, Rita Mancini, Gennaro Ciliberto

**Affiliations:** 1Preclinical Models and New Therapeutic Agents Unit, IRCCS Regina Elena National Cancer Institute, 00144 Rome, Italy; cirofrancescoruggier@libero.it (C.F.R.); salvati.sv@gmail.com (V.S.); 2Department of Experimental and Clinical Medicine, University “Magna Graecia” of Catanzaro, 88100 Catanzaro, Italy; malpicci.debora@gmail.com; 3Department of Molecular and Clinical Medicine, Laboratory Affiliated to Istituto Pasteur Italia Fondazione Cenci Bolognetti, University of Rome “Sapienza”, 00161 Rome, Italy; luigifattore1985@gmail.com; 4Department of Melanoma, Oncologic Immunotherapy and Innovative Therapies Istituto Nazionale Tumori IRCCS, “Fondazione G. Pascale”, 80131 Naples, Italy; g.madonna@istitutotumori.na.it (G.M.); v.vanella@istitutotumori.na.it (V.V.); d.mallardo@istitutotumori.na.it (D.M.); me.capone@istitutotumori.na.it (M.C.); p.ascierto@istitutotumori.na.it (P.A.A.); 5Department of Molecular and Clinical Medicine, University of Rome “Sapienza”, 00161 Rome, Italy; domenico.liguoro@uniroma1.it (D.L.); rita.mancini@uniroma1.it (R.M.); 6Department of Biochemistry, Case Western Reserve University, Cleveland, OH 44106, USA; bxb602@miami.edu; 7Department of Dermatology, Miller School of Medicine, Miami, FL 33136, USA; 8Scientific Directorate, IRCCS Regina Elena National Cancer Institute, 00144 Rome, Italy

**Keywords:** melanoma, target therapy, adaptive resistance, ErbB3 phosphorylation, CTCs, neuregulin1

## Abstract

In recent years the introduction of target therapies with BRAF and MEK inhibitors (MAPKi) and of immunotherapy with anti-CTLA-4 and anti-PD-1 monoclonal antibodies have dramatically improved survival of metastatic melanoma patients. Despite these changes drug resistance remains a major hurdle. Several mechanisms are at the basis of drug resistance. Particular attention has been devoted over the last years to unravel mechanisms at the basis of adaptive/non genetic resistance occurring in BRAF mutated melanomas upon treatment with to MAPKi. In this paper we focus on the involvement of activation of ErbB3 receptor following early exposure of melanoma cells to BRAF or MEK inhibitors, and the following induction of PI3K/AKT pathway. Although different mechanisms have been invoked in the past at the basis of this activation we show here with a combination of approaches that autocrine production of neuregulin by melanoma cells is a major factor responsible for ErbB3 phosphorylation and downstream AKT activation. Interestingly the kinetic of neuregulin production and of the ensuing ErbB3 phosphorylation is different in different melanoma cell lines which underscores the high degree of tumor heterogeneity. Moreover, heterogeneity is further highlighted by the evidence that in different cell lines neuregulin upregulation can occur at the transcriptional or at the post-transcritpional level. Finally we complement our study by showing with a liquid biopsy assay that circulating tumor cells (CTCs) from melanoma patients undergo upregulation of ErbB3 phosphorylation in vivo shortly after initiation of therapy.

## 1. Introduction

Malignant melanoma is the most aggressive tumour of the skin, which arises from the transformation of melanocytes. Its incidence has increased significantly in the last decades [1]. Over the past few years thanks to the development of immunotherapy with immune checkpoint inhibitors and targeted therapy against kinases of the RAS/BRAF/MAPK pathway, the clinical outcome has improved dramatically with the achievement of long term benefit in approximately 50% of patients with metastatic disease [2,3].

Approximately 50% of melanomas at diagnosis harbour mutations in the B-Raf proto-oncogene serine/threonine kinase (BRAF) gene, a key serine-threonine kinase in the MAPK signalling pathway. The most common mutation is a substitution of a valine to glutamic acid (V600E) which occurs in about 90% of cases, but also other mutations V600K-D-R may occur [4]. All these aminoacid substitutions result in constitutive kinase activation and uncontrolled cell growth. This led initially to the clinical development of BRAF inhibitors such as vemurafenib or dabrafenib with promising results in terms of high objective response rates, and improved survival [5]. However, the strong evidence that disease relapse due to drug resistance occurs shortly after initiation of BRAF inhibitor monotherapy, and that this is linked to the emergence of bypass mutations in resistant tumours which cause reactivation of the RAS/BRAF/MEK pathway [2], led to the development of dual therapies with BRAF and MEK inhibitors. Combo therapy with a BRAF and a MEK inhibitor has, therefore, become the current standard of care [6]. Unfortunately, however, also dual therapy, although being able to provide more durable disease control and improved survival vs. monotherapy, is plagued by the development of drug resistance [2,6].

Intense efforts have been directed to identify additional mechanisms of resistance, in particular adaptive non-genetic mechanism, which may help melanoma cells to survive the stress linked to the exposure to BRAF/MEK inhibitors. In this context, members of the EGF receptor family have emerged as promising targets in the treatment of various forms of cancer, due to their fundamental involvement in the activation of the proliferation and survival pathway induced by the PI3K / AKT pathway [7]. In particular the ErbB3 receptor has been identified as one of the key elements of malignancy, being, in fact, one of the most potent activator of the AKT pathway because of the presence of six tyrosine residues in its intracellular domain which when phosphorylated represent docking sites for proteins involved in proliferative and pro-survival signalling [8,9,10,11,12]. ErbB3 is frequently over-expressed in human melanoma cells [8,9]. Immuno-histochemical analysis showed high ErbB3 levels of expression in melanoma metastases and its association with disease progression [13,14]. ErbB3 is also involved in the development of resistance by tumour cells to conventional anti-EGFR and anti-ErbB2 therapies [15,16,17]. Furthermore, ErbB3 signalling induced by its natural ligand, neuregulin (NRG), is able to inhibit the differentiation of melanocytes, promoting the proliferation of melanoma cells [18]. Previous studies have shown that ErbB3 is a key factor in the development of adaptive resistance to BRAF and Mek inhibitors, albeit with different mechanisms, and that ErbB3 inhibition with monoclonal antibodies is able to potentiate the efficacy of target therapy in melanoma [8,15,19,20,21]. In this study we have provided further insights into the central role of ErbB3 and its mechanism of activation.

## 2. Results

### 2.1. BRAF-Mutated Melanoma Cell Lines Show Different Activation Kinetics of the pErbB3/pAKT Axis After Exposure to BRAF Inhibitors

In order to confirm if the main mechanism responsible for ErbB3 phosphorylation observed in melanoma cells exposed to BRAF inhibitors is the activation of an autocrine loop, we assessed the effect of the conditioned medium (CM) obtained from different cell lines after incubation with vemurafenib as a BRAF inhibitor. In detail three BRAF-mutated melanoma cell lines (WM266, WM115 and LOX IMVI) were treated with vemurafenib (0.5 µM) for different times (2 h, 8 h and 24 h). Of note we decided to use a dose of 0.5 μM BRAFi for western blotting experiments because different cell lines have different degrees of sensitivity to BRAFi and this drug concentration is able to inhibit cell growth efficiently in all cell lines used [15,19]. The conditioned medium was collected and added to starved melanoma cells for 30 min (Figure 1A). In parallel melanoma cell cultures from the same cell lines were directly exposed to vemurafenib. Cell extracts were then subjected to Western Blot analysis (Figure 1B and Appendix A). The results show a strong phosphorylation of ErbB3 in melanoma cells upon treatment with CM from BRAF inhibitor-exposed melanoma cells, similar to what happens after direct cell exposure to the same BRAF inhibitor. Of note, increase of pErbB3, which is always accompanied by an increase of pAKT levels, occurs with different kinetics in the three cells lines: in WM266 is already high at 2 h, remains constant at 8 h and declines thereafter; in WM115 is low at 2 h, maximal at 8 h and sharply declines after 24 h; in LOX IMVI is high only after 24 h. These findings support the notion that melanoma cells respond to BRAF inhibitors exposure by releasing in the culture media one or more soluble factor able to activate the ErbB3/AKT axis.

### 2.2. Early Release of NRG-1 Is Responsible for the Activation of the ErbB3 Receptor and NRG-1 Inhibition Potentiates the Growth Inhibitory Effect of BRAF Inhibition

In order to assess the involvement of the ErbB3 ligand neuregulin-1 (NRG-1) we evaluated the levels of secreted NRG-1 in WM266 cell medium by immunoassay experiments (see Materials and Methods section) (Figure 2A and Appendix A).

The results show that exposure to BRAF inhibitor is able to induce a strong and fast increase of secreted NRG-1 in WM266-derived cell medium. In particular, a high amount of neuregulin-1 (more than 10-fold) was detected shortly after drug exposure which gradually declined at later time points. Furthermore, to confirm that phosphorylation of ErbB3 receptor is a consequence of increased NRG-1 production by melanoma cells we decided first to treat WM266 melanoma cells with vemurafenib (0.5 µM) or not for 24 h. Then CM was collected and pre-incubated with a neutralizing antibody against NRG-1 (Anti-NRG1) for 1h before incubation with starved WM266 cells. Western blotting results clearly show that Anti-NRG1 completely abrogated the phosphorylation of ErbB3 receptor and the activation of the PI-3K/pAKT pathway (Figure 2B and Appendix A). Moreover, viability assays showed that the Anti-NRG1 enhanced the inhibitory effect of BRAFi on WM266 cell growth especially at lower drug doses (Figure 2C,D). These data have been confirmed in colony formation assays both in WM266 and WM115 cells as reported in Appendix A. As a control a non-neutralizing Anti-ErbB3 antibody (A2) [15] was unable to exert an inhibitory effect on cell proliferation (Figure 2D).

### 2.3. BRAF-Mutated Melanoma Cells Show Different Patterns of Changes in ErbB3, NRG-1 and FOXD3 Gene Expression Levels after BRAFi Treatment

To better investigate the different kinetics of ErbB3 activation in the three different cells lines, WM266, WM115 and LOX IMVI were exposed to vemurafenib from 2 to 72 h, cell extracts were prepared and subjected to western blotting. Results demonstrated that phospho-ErbB3 follows similar pattern in all three lines, with an average peak of activation at 12/24 h. It is important to point out that only in WM266 cells there is a slight and earlier activation of the receptor before 12 h. As to the total ErbB3 protein, it is stable until 48 h in WM266 and WM115 cells following BRAF inhibition. Differently, in LOX IMVI cells ErbB3 protein increases with a peak at 12 h and then decreases in its expression levels from 24 to 72 h (Figure 3A and Appendix A).

In order to further assess changes in the expression levels of NRG-1, ErbB3 and Forkhead box D3 (FOXD3) genes, quantitative RT-PCR analysis was carried out on total RNA extracted from cells exposed to vemurafenib at different times (Appendix A). FOXD3 was chosen since it is a known regulator of ErbB3 gene expression [22]. The results are diagrammatically summarized in Figure 2B using a color coded scheme (Figure 3B). In synthesis we observed the existence of three different patterns of changes in ErbB3, NRG-1 and FOXD3 gene expression levels over time: (1) in WM266, NRG-1 and FOXD3 mRNA levels increase early after drug exposure (2 h) followed by an increase of ErbB3 mRNA at a later times (48–72 h); (2) in WM115 a strong increase of NRG-1 and FOXD3 mRNA levels occurs at intermediate times (12–24 h), which still precedes the increase of ErbB3 expression (48–72h); (3) finally in LOX IMVI, M14 and A375 we observed a late and simultaneous increase (48–72 h) of the expression levels of all three genes.

While these data underscore the strong degree of heterogeneity of melanoma cell lines, they confirm that one of the prominent and constant adaptive response is the upregulation of NRG-1 expression. Interestingly in none of the three cells lines NRG-1 upregulation follows temporally FOXD3 upregulation. Although we have not carried out gene silencing studies of FOXD3, based on the kinetics of expression, we tend to exclude therefore that FOXD3 is involved in induction of NRG-1 expression.

### 2.4. NRG-1 Activation Occurs via Different Mechanisms

Based on previous results we decided to focus our attention on two melanoma cell lines, namely WM266 and WM115, which show early and intermediate times of NRG-1 up-regulation upon BRAF inhibition respectively. In particular, we decided to investigate whether the increase of NRG-1 mRNA could be related to increased gene transcription. To this purpose we used a human NRG1 promoter-luciferase reporter construct (pGL3-NRG1) previously described by Zhang et al. [23]. pGL3-NRG1 contains two distinct promoter regions from nucleotide −3987 to −3644 and from nucleotide −777 to +193 relative to the CAP site and is dependent upon Notch activation. Melanoma cells were transfected with pGL3-NRG1, exposed or not to vemurafenib. Cell extracts were prepared and assayed for luciferase activity. The results (Figure 4A) show that while in WM115 promoter activity is increased several-fold after BRAF inhibitor treatment (Figure 4A right panel), in WM266 cells there is no drug-induced promoter activation (Figure 4A left panel). Moreover the same experiments were performed also in A375 cells which have not shown a significant increased promoter activity after BRAF inhibitor treatment as observed in WM266 cell line (Appendix A). This result has been observed in at least three separate experiments. These findings support the notion that NRG-1 activation may be due to different mechanisms in the two cell lines, i.e., transcriptional in WM115 and post-transcriptional in WM266. To confirm this WM115 and WM266 cells were exposed to vemurafenib as BRAF inhibitor and co-treated with actinomycin D, a known inhibitor of RNA polymerase. Total RNA was extracted 24 h after BRAFi exposure. Quantitative RT-PCR analysis were performed on NRG-1 and ErbB3 mRNAs. This confirmed (Figure 4B) that after treatment for 24 h with BRAF inhibitor alone, NRG-1 but not ErbB3 was strongly induced in both WM266 and in WM115. When WM115 cells were co-treated with BRAF inhibitor and actinomicin D, upregulation of NRG-1 mRNA levels was totally inhibited (Figure 4B right panel). This result is in line with the upregulation of promoter-driven luciferase expression shown above and confirms the notion that in this cell line increased neuregulin expression is under transcriptional control. In contrast, in WM266 cells NRG-1 mRNA levels were not decreased by simultaneous exposure to Actinomycin D (Figure 4B left panel). This result suggests that in this cell line NRG-1 upregulation following drug exposure is not under transcriptional control.

### 2.5. pErbB3 Upregulation Is Observed in Circulating Melanoma Cells After Treatment with MAPK Inhibitors

In order to assess if neuregulin-induced pErbB3 activation occurs also in the clinical setting we decided to carry out a clinical study in BRAF mutated melanoma patients undergoing standard dual therapy with MAPK inhibitors. To this purpose we focused our attention on Circulating Tumor Cells (CTCs). CTCs are cells that derive from the tumour bulk but are capable of spreading throughout the body because after the detachment from primary tumour sites, are able to resist programmed cell death (a phenomenon also known as “anoikis”). Evidence of the biological and clinical significance of CTCs is rapidly increasing. The use of the standardized systems to detect rare tumour cells in the blood of patients with a variety of tumours has facilitated the enumeration of CTCs and their monitoring over time [24,25,26]. Furthermore, fluorescent staining of isolated CTCs may allow some level of functional analysis. In recent years the study of CTCs in melanoma has been proposed as tool for the surveillance of metastatic melanoma patients [27] and has been also shown to be helpful to better understand clonal heterogeneity in melanoma [28].

In the present study we enrolled a total of 11 BRAF-mutated metastatic melanoma patients. Patients characteristics are reported in Appendix A. Blood samples were collected before initiation of therapy with BRAF and MEK inhibitors and after short term. Samples were processed as described in the materials and methods section and CTCs were collected on membrane filters (Figure 5A). Filters were subjected to an immunofluorescence assay using a specific staining to identify the expression of the phosphorylated ErbB3 receptor (Figure 5B and Appendix A). Further images in which the activation of pErbB3 was detected in CTCs of melanoma patients after MAPKi treatment are reported in Appendix A. A technical note about these data: both DAPI dye and antibody used to stain cell nuclei and pErbB3 respectively, can remain trapped in the pores causing their coloring by autofluorescence. In addition since leucocytes can be present in the filter because their size is larger than that of the pores a specific staining for CD45 was performed in order to distinguish them from CTCs (Appendix A). In our samples we observed a very low proportion of white cells. As expected CTCs not showed the expression of CD45 receptor as compared to leucocytes (Appendix A) [29,30].

Results are reported in Figure 6 and Appendix A. They show that in BRAF-mutated melanoma patients after 3 days of treatment with BRAF and MEK inhibitors there is a statistically significant increase in the number of CTCs bearing activated pErbB3 receptor as compared to CTCs before initiation of treatment (Figure 6A). Of note, in some cases, the activation of the pErbB3 receptor was observed only after the treatment. In addition, in eight patients out of 11 we observed an increased activation of ErbB3 receptor (Figure 6B). Data were also used to plot the Receiver Operating Characteristic (ROC) curve (Figure 6C) which indicates that ErbB3 phosphorylation describes the treatment with MAPK inhibitors with a significant AUC of 0.70.

## 3. Discussion

The NRG1/ErbB3/AKT axis has been shown to be involved in adaptive drug resistance in several tumour types, including, breast, lung, prostate, cancer and melanoma [11,12,16,31,32,33,34,35]. Cancer cells of different origin, when exposed to standard chemo- or target-therapy promptly react through the activation of a common survival pathway centred around the EGFR receptor family member ErbB3 and its downstream signalling elements PI3K and AKT. The high frequency with which this particular pathway is activated in adaptive resistance is justified by its intrinsically high transducing capacity which is due to the strong processivity of the kinase domain of ErbB3 common dimerization partner HER2/ErbB2 and to the presence in the intracytoplasmic domain of ErBB3 of several Tyr residues substrates for phosphorylation as docking sites for PI3K (ErbB3) [18].

The most common ligand of the ErbB2/ErbB3 heterodimer is NRG1 despite several other members of the Neuregulin family have been found to be involved in its activation. These proteins derive from the alternative splicing of four genes giving rise to at least 26 different isoforms [36]. However NRG1 is described as the major ligand involved in drug resistance mediated by ErbB2/ErbB3 heterodimer activation. A group of studies, mostly in melanoma but also in breast and thyroid cancer, linked NRG1-induced ErbB3 activation to a paracrine effect and to a remodelling of the tumour microenvironment following exposure to BRAF and MEK inhibitors involving stromal fibroblasts as the main source of NRG-1 mediated PI3k/AKT activation [4,20,37,38]. In other cases NRG1 was shown to be produced by the same tumour cells and to induce ErbB3 phosphorylation via an autocrine lop [16]. We have previously proposed this last mechanism to be active also in melanoma [19]. In order to assess in greater detail this aspect we conducted the present study in a panel of V600 BRAF-mutated human melanoma cell lines and utilizing a combination of approaches involving the use of conditioned medium from BRAFi exposed cells and of neutralizing antibodies against NRG1. All these approaches confirmed the validity of the autocrine loop model, the most compelling proof of this was the demonstration that antibodies against NRG1 synergize with a BRAFi in the inhibition of melanoma cell growth in the absence of exogenously added NRG1. In summary we believe that at least in melanoma, when cells are exposed to targeted therapy, redundant cell autonomous and non- cell autonomous mechanisms are activated leading to the production of NRG1 by both melanoma cells and cells of the tumour microenvironment which converge into the activation of the ErbB3/PI3K/AKT pathway.

It has been previously shown that the transcription factor FOXD3 is rapidly induced in melanoma cells upon inhibition of MAPK signalling and that this phenomenon contributes to adaptive resistance to RAF inhibitors also in part through enhanced transcription of the ErbB3 receptor gene [22]. More recently FOXD3 activation has been attributed to sumoylation of transcription factor SOX10 which is inhibited by ERK activity [39]. Interestingly our data confirm in all BRAF mutated melanoma cell lines analysed that FOXD3 mRNA is upregulated, however it never precedes temporally upregulation of NRG1, which, to our opinion, remains therefore the principal driver of adaptive resistance to BRAF and MEK inhibitors.

Our study reveals that a variety of mechanisms can be a basis of NRG1 upregulation. In support of this concept we demonstrated thanks to a combination of transfection experiments with a hybrid promoter construct and of transcription inhibition assays that, while in WM115 cells NRG1 is transcriptionally upregulated, in WM266 a post-transcriptional mechanism must be at play. We have been unable so far to identify this last mechanism. We can only exclude at this time that this is due to increased mRNA stability consequent to decreased levels of miRNA-548 which was postulated through TargetScanHuman bioinformatic analysis (http://www.targetscan.org/vert_72/) [40] as a potential binder of the NRG1 mRNA 3′ untranslated region. However, these are not the only possible mechanisms that can be responsible for NRG1 upregulation. Indeed Ebbing et al. previously showed in a gastroesophegeal cancer model that resistance to trastuzumab was caused by increased release of surface heregulin following upregulation of metalloprotease ADAM10 [41]. With regard to the transcriptional activation observed in WM115 cells following cell exposure to MAPK inhibitors, a potential explanation could be the activation of Notch pathway in which factor Notch1 directly regulates the transcription of neuregulin by binding to its promoter region contributing in this way to melanoma tumorigenicity [23,42,43,44].

The involvement of ErbB3 in the development of drug resistance has been shown so far only in in vitro or in vivo pre-clinical models. A demonstration that this mechanism could be active also in the clinic had been missing. We have tried to fill this gap of knowledge by addressing the question of whether increased ErbB3 phosphorylation takes place also in patients soon after initiation of therapy with MAPK inhibitors. We decided to follow a non-invasive approach directed to analyse by immunofluorescence the phosphorylation status of the ErbB3 receptor in CTCs isolated from BRAF mutated melanoma patients before the initiation of therapy and 3 days after the start of therapy. The results, albeit obtained from a small number (*n* = 11) of patients, show a statistically significant increase of ErbB3 specific phosphorylation in circulating melanoma cells. In other types of tumours such as ovarian cancer and glioblastoma the expression of ErbB3 receptor had been detected before in CTCs [24]. For the first time we identified the presence of phospho-ErbB3 receptor in CTCs isolated from human melanoma plasma samples. Our data suggest that also melanomas activate the ErbB3 receptor soon after exposure to MAPK inhibitors probably through NRG1 overexpression in order to counteract the anti-proliferative effect of targeted therapy. Activation of ErbB3 receptor in CTCs in response to MAPK inhibitor treatments, could help promoting survival and metastatization of circulating malignant melanoma in other locations of the body.

## 4. Materials and Methods

### 4.1. Cell Lines and Treatments

Human melanoma cell lines, WM266, LOX IMVI, M14, A375 and WM115 were cultured in RPMI supplemented with 10% FBS. To evaluate ErbB3, AKT and ERK 1/2 signalling melanoma cells were seeded and the following day they were serum starved overnight. Afterwards cells were exposed to vemurafenib for the indicated time points (2, 8 and 24 h). Conditioned medium (CM) was collected at each time point and transferred to treatment naive serum starved melanoma cells for 30 min. Differently melanoma cells were also treated from 2 h to 72 h in order to study the activation kinetics of the ErbB3/Akt axis by western blotting assay. BRAF-mutated human melanoma cell lines, LOX IMVI, A375, M14 (V600E), WM115, WM266 (V600D) were obtained from the laboratory of Dr. Paolo A. Ascierto at National Cancer Institute of Naples “Fondazione G. Pascale”, Naples, Italy.

### 4.2. Antibodies and Reagents

Antibodies for western blotting analyses were purchased from Cell Signaling Technology (Boston, MA, USA). Anti-ErbB3 and anti-GAPDH were obtained from Santa Cruz Biotechnology (Dallas, TX, USA) [13,17]. Anti-rabbit and anti-mouse were purchased from AbCam (Cambridge, UK). Vemurafenib and trametinib were obtained from Selleck Chemicals (Houston, TX, USA). TaqMan probes for ErbB3, HRG, FOXD3 and housekeeping gene GAPDH were purchased from Applied Biosystems (Foster City, CA, USA). Actinomycin D was purchased from Sigma (St. Louis, MO, USA). The anti-NRG (blocking peptide) was purchased from Thermo Fisher Scientific (Waltham, MA, USA). Immunofluorescence primary antibody phospho-ErbB3 was obtained from Santa Cruz Biotechnology.

### 4.3. NRG-1 Detection

Released NRG-1 in cell culture media was detected by an immunoassay through Bio-Plex system Bio-Rad (Hercules, CA, USA) according to manufacturer’s instructions. Briefly, laser excitation is used to determine NRG-1 concentration by measuring the reporter dye fluorescence and fluorescence signal is in direct proportion to the amount of NRG-1 bound.

### 4.4. Western Blot Analysis

Melanoma cells were lysed with RIPA buffer; 50 μg of total protein were resolved under reducing conditions by 8% SDS-PAGE [45]. The membranes were blocked with 5% non fat dry milk in PBS 0.1% Tween 20, and incubated with the different primary antibodies. GAPDH and Vinculin were used to estimate the protein equal loading. Densitometric analysis was performed using Quantity One Program (Hercules, CA, USA).

### 4.5. RNA Extraction and Real-Time PCR Analysis

RNA was extracted using TRIzol method (Thermo Fisher Scientific) [46] according to manufacturer’s instruction. Total RNA was quantitated by spectrophotometry. Real Time-PCR was assayed by TaqMan Gene Expression Assays (Applied Biosystems). Each targeted transcript was validated using the comparative Ct method for relative quantification (ΔΔCt) reference to the amount of a common reference gene (GAPDH) [46].

### 4.6. Luciferase Assays

The NRG1-reporter plasmid was provided by B. Bedogni [23]. Melanoma cells were transfected using the Lipofectamine 2000 (Thermo Fisher Scientific) according to the manufacturer’s instructions. After 48 h, cells were lysed in 100 μL lysis buffer (Promega, Madison, WI, USA). Activities of Firefly and Renilla were assessed by the dual-luciferase Assay system (Promega) and light production was measured for 10 s in GloMax-Multi Detection System (Promega).

### 4.7. Cell Viability

The number of viable melanoma cells was mesured by quantification of the ATP present according to CellTiter-Glo^®^ Luminescent Cell Viability assay protocol. WM266 melanoma cells were seeded in a 96 multiwell plate and trated with different doses of BRAF inhibitor together to peptide anti-NRG1 (the inefficient A2 antibody against ErbB3 was used as control). Colony formation assays have been performed as previously described [47,48].

### 4.8. Patient Selection

The use of human samples was approved by Istituto Pascale’s Ethical Committee with the protocol DSC/1504 on June 11, 2014. Patients eligible for inclusion in this study have all the following criteria:(1)Patients ≥18 years old.(2)Patients able to understand and willing to sign the informed consent form and able to adhere to the study visit scheme and other protocol requirements. Written informed consent must be obtained before any procedure.(3)Diagnosis confirmed by locally advanced stage melanoma histology or metastatic (stage IIIB–IV according to the American Joint Committee on Cancer—AJCC).(4)The patient is a candidate for therapy with BRAF inhibitors and/or MEK inhibitors.

### 4.9. Circulating Tumor Cells (CTCs) Isolation

ScreenCell^®^ size exclusion technology was used (see Figure 5A) to isolate circulating tumor cells. In particular melanoma patients’ blood samples were collected in a K2-EDTA tubes Patients’ blood samples were collected in aK2-EDTA tubes. The screenCell FC2 buffer was prepared bringing it to the optimal pH (pH between 6.7 and 7) through the use of NaOH. Then 3 mL of blood and 4 mL of screenCell FC2 buffer were added into a sterile 15 mL tube and mixed together by inverting the tube. Incubate for 8 min at room temperature. At this point the solution was located in the upper part of the ScreenCell cyto device which in detail consists of a column (module A) and a vacuum tube (module B) inserted in the lower part (module C). The negative pressure released will allow the passage of solution through the filter located between the lower part of the column (module A) and the module B. The filtered blood was collected in the vacuum tube (module C). When the solution (blood + buffer) was about to end, 1.6 mL of PBS was added in order to wash the filter. At the end of the process the filter was recovered leaving it to dry on absorbent paper for an hour and then processed by immunofluorescence or stored at −20 °C.

### 4.10. Immunofluorescence

Each filter was rinsed with 100 microliters of PBS for 10 min. Later 100 microliters of the “blocking solution” composed of 10% BSA, 100% Triton and 100% FBS were used for 1h at room temperature. Then two more PBS washes were made again before using a 10% BSA solution, 100% Triton where the primary antibody is diluted 1:50. Of the following solution 100 microliters per well were used for 1 h at 37 °C. After that a couple of PBS washes were done, the secondary antibody was used for 40 min at 4 °C. Finally, nuclear staining was performed with the DAPI diluted 1:1000 for 5 min at room temperature before reading with the fluorescence.

### 4.11. Statistical Analyses

All results reported in this manuscript are presented as mean values from three independent experiments ± S.D. ROC curves have been plotted as previously described [47]. Quantitative analysis for curve fitting have been performed through KaleidaGraph software (version 4.1; Synergy Software, Reading, PA, USA) [19].

## 5. Conclusions

Our study further underscores the importance of non-mutational adaptive mechanism of resistance to target therapy in metastatic melanoma. This has significant implications for the development of more powerful combination therapies capable to provide a longer-term control of disease. In the last year a plethora of promising non mutational mechanisms have been discovered which involve both intracellular proteins [2,49] and non-coding RNAs [47,48,50,51,52] which are waiting for testing in the clinical setting.

The development of biological assays such as the one we have shown here in CTCs may help to guide the development of new therapeutic combinations. In fact, our new findings could have potential diagnostic implications in the field of liquid biopsy, as the activation of the ErbB3-NRG1 axis could be implicated as a new early biomarker able to predict the response to the targeted therapies in metastatic melanoma patients. It will be interesting in the future to confirm these data in a larger cohort of patients and to establish a possible correlation between early (and perhaps more pronounced) activation of pErbB3 in melanoma CTCs and clinical outcome.

## Figures and Tables

**Figure 1 cancers-11-01425-f001:**
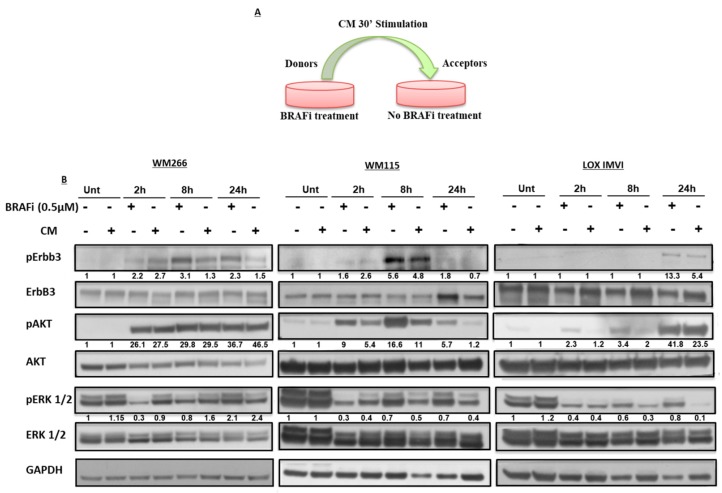
The activation of pErbB3/pAKT survival pathway occurs in different times upon BRAFi exposure in BRAF-mutated melanoma cell lines. (**A**) In the schematic representation of the experiment unstimulated melanoma cells were exposed for 30 min with conditioned medium (CM) coming from BRAF inhibitor-stimulated melanoma cells. (**B**) WM266 (left panel), WM115 (middle panel) and LOX IMVI (right panel) melanoma cell lines were starved and treated with BRAFi (0.5 μM) for 2 h, 8 h and 24 h. pErbB3/ErbB3, pERK/ERK and pAkt/Akt values are expressed as fold change with respect to the control unstimulated cells expression levels by densitometric analysis. Conditioned medium from BRAF inhibitor-stimulated melanoma cells induce an early pErbB3/pAKT axis activation as shown in western blotting experiments.

**Figure 2 cancers-11-01425-f002:**
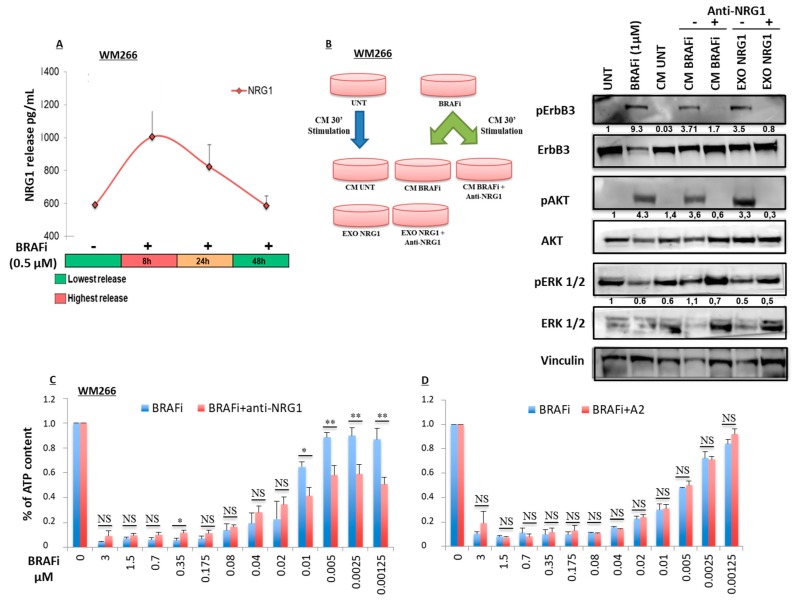
The inhibition of released NRG-1 enhanced the inhibitory effect of BRAFi on WM266 cell growth. (**A**) WM266 melanoma cell lines were starved and then treated with BRAFi (0.5 μM) for different times (8 h, 24 h and 48 h). Cell media was collected and analyzed by immunoassay using the NRG-1 capture antibody. Results shows that increasing levels of NRG-1 are released in cell media upon short term BRAF inhibitor treatment in WM266 melanoma cells. (**B**) In the schematic representation of the experiment WM266 cell line was serum starved for 24 h and later treated or not with BRAFi (1 μM) for other 24 h before collecting its culture medium or also called conditioned media (CM). Then the CM were subsequently pre-incubated or not with the anti-NRG1 antibody for 1h. WM266 cells treated for 1 h with the untreated conditioned medium (untreated CM) or BRAFi-stimulated (BRAFi CM) were pre-incubated or not with exogenus neuregulin-1 (EXO NRG1). The treatment with the anti-HRG antibody completely abrogates both ErbB3 and AKT phosphorylation induced by BRAFi CM and EXO NRG1 through western blotting assay. The expression of pErbB3/ ErbB3, pERK/ERK and pAKT/ATK was evaluated by densitometric analysis and expressed as fold change with respect to the control unstimulated cells to which value = 1 was assigned. (**C**) Anti-NRG1 blocking peptide improved the inhibitory effect of BRAFi on WM266 cell growth especially at lower drug doses as evidenced by a lower ATP release. (**D**) Conversely the use of a non-efficient anti ErbB3 antibody (A2) used as control [15], did not enhances the inhibitory effect of BRAFi on cell growth. *p* value was calculated using the T-test whose significance is expressed as *p* < 0.05. *: *p* < 0.05, **: *p* < 0.01, NS: *p* > 0.05.

**Figure 3 cancers-11-01425-f003:**
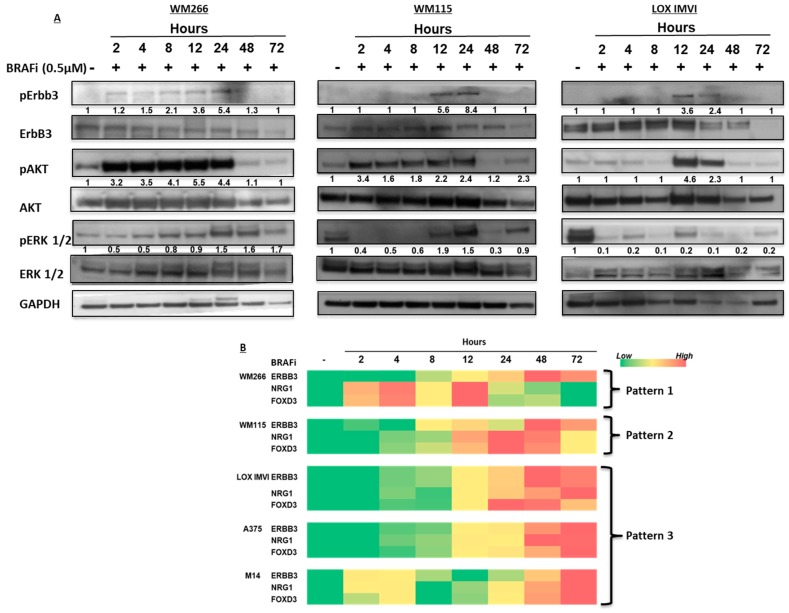
Different kinetics of ErbB3 activation occur in BRAF-mutated melanoma cell lines. (**A**) WM266 (left panel), WM115 (middle panel) and LOX IMVI (right panel) melanoma cells were starved and then treated with BRAFi (0.5 μM) for different times (from 2 h to 72 h). ErbB3/Akt axis is early activated after BRAF inhibitor treatments with a different manner. Densitometric analysis values were used to evaluate pErbB3/ErbB3, pERK/ERK and pAkt/Akt expression with respect to the control untreated cells. (**B**) In the same experimantal condition (panel A) also total RNA was extracted and subjected to qRT-PCR in order to evaluate the kinetic activation of of NRG1, ErbB3 and FOXD3 genes. Results are represented as a “heat map”. Colors in the heat map indicate no increase in gene expression (green); moderate increase (yellow) or a strong increase (red), respectively. GAPDH reference gene was used for normalization.

**Figure 4 cancers-11-01425-f004:**
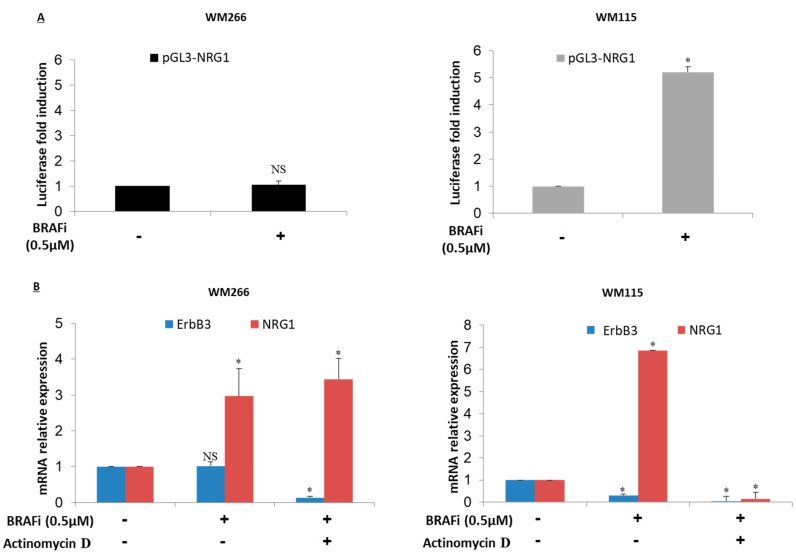
WM266 and WM115 melanoma cells show different mechanisms of NRG-1 activation. (**A**) WM266 and WM115 cells were transfected with NRG1-reporter plasmid and treated or not with BRAF inhibitor to 0.5 µM for 24 h. The luciferase gene was cloned downstream of NRG1. The luciferase activity was evaluated by the Renilla Luciferase assay. The results show that NRG1 gene promoter activity is enhanced after BRAF inhibitor treatment only in WM115 cells while no luciferase induction occurred in WM266 cell line. (**B**) Melanoma cells were co-treated with BRAF inhibitor (0.5 µM) and actinomicin D transcription inhibitor (5 µg/mL) for 24 h and subsequently collected and subjected to qRT-PCR analysis to evaluate ErbB3 and NRG1 gene expression. Results show that only in WM115 cells NRG1 expression is under transcriptional control. GAPDH reference gene was used for normalization. *: *p* < 0.05. NS: *p* > 0.05.

**Figure 5 cancers-11-01425-f005:**
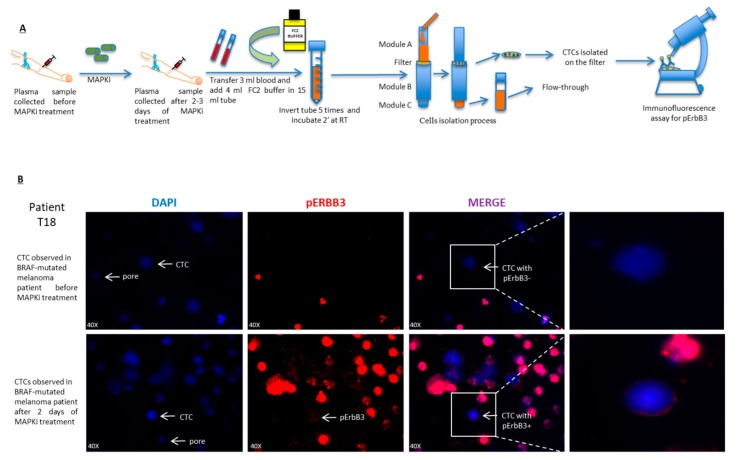
ErbB3 activation is observed in circulating melanoma cells after short term treatment with MAPK inhibitors. (**A**) Schematic representation A shows all steps to isolate the circulating tumor cells (CTCs) from plasma samples of the 11 metastatic melanoma patients before and after short term treatment with MAPK inhibitors. (**B**) CTCs isolated by CELLSEARCH^®^ Circulating Tumor Cell Kit are subjected to Immunofluorescence analysis in order to evaluate pErbB3 activation through pimary pErbB3 antibody and secondary Anti-Texas Red Antibody. T18 melanoma patient shows the activation of phospho-ErbB3 receptor after 2 days of MAPKi treatment in circulating tumor cells (CTC with pErbB3+) (Dapi = CTCs nuclei; Red = pErbB3). Scale bars: 50 μm.

**Figure 6 cancers-11-01425-f006:**
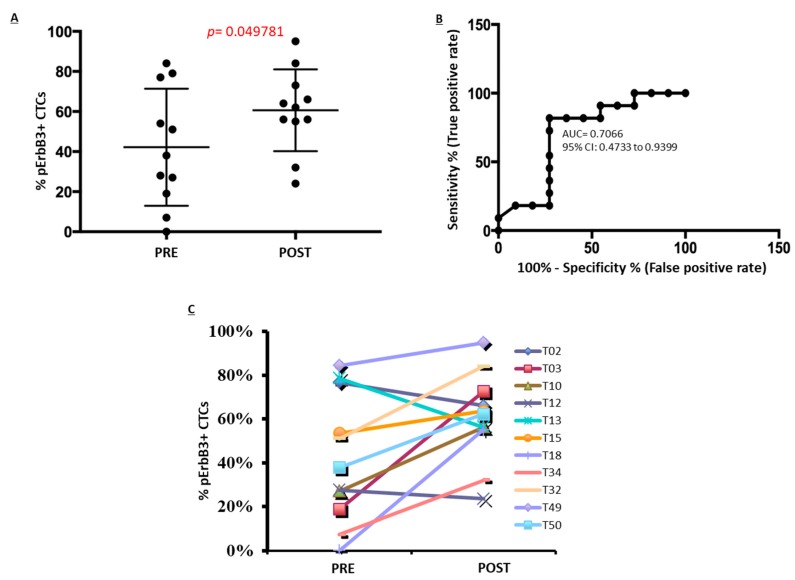
Statistical analysis. (**A**) Box-plot graph was obtained by GraphPad Prism software. Results show that the percentage of phospho-ErbB3 results to be activated greater in CTC’s melanoma patients upon treatment with MAPKi (*p* < 0.05). *p* value was calculated using the *t*-test whose significance is expressed as *p* < 0.05. (**B**) The second panel depicts the individual patients’ analysis to evaluate ErbB3 receptor activation before and after short term MAPK inhibitor treatment. (**C**) Receiver operating characteristic (ROC) curves estimating the predictive value of ErbB3 activation as early marker upon treatment with MAPK inhibitors.

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
