# Peer review of "ErbB3 Phosphorylation as Central Event in Adaptive Resistance to Targeted Therapy in Metastatic Melanoma: Early Detection in CTCs during Therapy and Insights into Regulation by Autocrine Neuregulin"

_cancers, 2019, doi:10.3390/cancers11101425_

Round 1

Reviewer 1 Report

The authors present new evidences for a central role of ErbB3 phosphorylation during the adaptive resistance to BRAF inhibition in melanoma via an autocrine loop mediated by NRG1 secretion. In addition, they present ErbB3 phosphorylation as an early event which can be measured in Circulating Tumor Cells in patient’s blood and be potentially used as a predictive biomarker to targeted therapy.

Although the conclusions of the first message are clear and supported by the results, the second message (CTC assay) looks weak and requires additional experiments to confirm the expected effect.

Overall, the switch from one cell line to another according to what we are looking at, is disturbing. It would be clearer if all 3 lines are used in all figures.

Major revisions need to be performed

MAJOR comments:

Figure 1:

How long were the cells maintained before conditioned medium collection? How long lasted the starvation? L94: the sentence should be more moderate. The words “clearly” and “strong” may be too strong. ErbB3 phosphorylation is only strong in WM115. Fig 1b: It would be appreciated to include a time point 12H for the kinetic of BRAFi on phospho ErbB3. If we relate these results to these in Fig 3: in WM115, the phosphoErbB3 pic of activation is at 8h in Fig 1, but at 12h/24h in Fig 3. Also in LOX IMVI, the pic of activation if at 24h in Fig 1 (there is no 12h to compare) but already at 12h in Fig 3.

 Can the authors explain this discrepancy?

Figure 2 :

L125: the word “proliferation” is somehow misleading. ATP is measured to estimate cellular metabolic activity (usually correlated to viability/ cell number). It is not a proliferation assay per se. Actually, the results of Fig 2C/D would strengenth a lot by an additional functional assay such as colony formation assay or “real” proliferation assay (EdU/ BrdU or a CFSE cell proliferation assay) Additionnally, these experiments would need to be repeated in at least a second cell line like WM115 where the activation of ERBB3 is stronger (according to Fig 1B). L118 : the kinetik of released NRG1 does not correlate to NRG1 RNA expression of heatmap in Fig 3. Actually, protein release (pic at 8h) appears before RNA (pic at 12h). Fig 2B: is the condition BRAFi + anti NRG1 missing? This condition would show us if BRAFi stimulation for 24h effectively leads to an autocrine loop of NRG1. In addition, wouldn’t we expect a synergistic effect (increase) on phosphoERBB3 under CM+BRAFi compared to BRAFi alone? Fig 2c: the major differences in ATP content are starting at 10nM and lower, it is therefore surprising to also see an effect at 0.35uM. Could the authors explain this fact?

Figure 3 :

According to Fig 2, the effect of BRAFi (via NRG 1 release) on ATP content is starting at 10nM and lower, therefore why every other experiments have been done with 0.5uM BRAFi treatment? This should be discussed in detail. Fig 3b: It is not clear what values were used to draw the heatmap. Increase or decrease compared to what ? If the reference was untreated condition, like in the fig S5, the whole first column of the heatmap should have the same color. Please explain better. If we compare RNA and protein expression in Figure 3, they are not always correlated. Example: WM266 untreated has high ERbB3 protein but low RNA. Also, ErbB3 protein decreases until 72h but RNA increases until the pic at 48h (opposite) In WM115, ErbB 3 protein is stable until 72h but RNA increases with pic at 48h LOXIMVI: why is there a protein doublet of ErbB3? This protein increases until a pic at 12h but rna starts medium, decreases early and then increases until pic at 48h The text L152 - 156 does not correctly describe the shown results. Phospho ErbB3 follows similar pattern in all 3 lines with a pic at 12H/24h (except in WM266 where it starts earlier). Suddently, 2 extra cell lines appear (A375 and M14). For which purpose ?

Figure 5 :

Although this CTC kit has been already described, it is based on physical properties of the circulating cells (i.e. size exclusion). Therefore it is highly recommended to confirm the characterization of the filtered cells as CTC (genomic or transcriptomic profiling for example, to check if they are tumor cells at all). How can we exclude the presence of contaminating blood cells ? Could they be counterstained in the immunostaining with a known leucocyte marker for instance? On which basis is the disctintion between pore and actual cell made with the DAPI staining ? Attention, the time of treatment in the patients is confusing. L237 states 72h but in the figure 5a, it is stated 2-3d and in the Fig 5b: 2d. L250 - 255: confusing description of results on erb activation. Are the authors describing the expression of ERBB3 or its activation ? The staining of phosphoERBB3 should always be compared to a total ERBB3 antibody staining to describe the actual activation of the protein. The immunofluorescence of phosphoERBB3 in the zoomed in cell in the treated condition does not show a positive staining. Why do the authors claim otherwise ?

Figure 6 :

Again, Fig6a is based on the analysis if the previous immunofluorescence staining (Fig 5b) which leads to serious doubts about what cells are we looking at and are they really positive for phosphoErbB3 ? Fig 6c and line 254: 3 out of 11 (30%) patients show a decrease in phosphoErbB3 signal after treatment, what does that mean for the accuracy of the prediction? In the text, the autors talk about « kinase inhibitor » treatment, because 2 different combinations were given to the patients. The 2 combination treatments should be mentioned in the text (or in the methods). According to Table S1, 4 patients had previous treatment with check point inhibitors. Since it is most likely that immunotherapy impacts the patients’ response to additional targeted therapy, the authors should discuss that there might be a biais in the patient cohort’s selection.

Discussion

ErbB2/ErbB3 heterodimer binds to other ligands than NRG1. Should it be discussed which ones and their potential role in the BRAFi resistance ?

L287 - 289: if this finding is the most important as the author claim, it should probably be backed-up by another functional assay in addition to ATP measurement (as already mentionned)

L289-292 : the central role of ErbB3 in this autocrine loop could be tested by gene silencing under the CM + anti-NRG 1 condition.

L327-328: It is still difficult to claim this, based on the data of figure 5 and 6.

Suggestion to enhance the clinical relevance of the data :

Could the author verify a potential correlation between erbB3 activity and melanoma patient survival (if not already done) ?

The melanoma patient cohort of the TCGA provides survival data on 470 patients (https://portal.gdc.cancer.gov/) and proteomic data (phosphoErbB3) from TCGA seem to be available according to this reference ( Li J, Lu Y, Akbani R, Ju Z, Roebuck PL, Liu W, et al. TCPA: a resource for cancer functional proteomics data. Nat Methods. 2013;10:1046–7.

As an alternative, a study on melanoma patients with BRAFi+MEKi and RNAseq data available (such as GSE50509 or more recent) would provide the expression status of a panel of Erbb3 transcriptional target genes (identified here : Mahmoud El Maassarani et al ; PLOS one 2016)

Minor comments:

Supplementals

Fig s2: statistical analysis should be provided

Fig s4: wm266: attention, the blot of gapdh does not correspond to the one in Fig 3a. Which one is the correct loading control ? this is quite important because if the correct loading control is that on Fig S4, then it is unclear if ErbB3 is phosphorylated at all under BRAFi treatment.

Fig s5: statistical analysis should be provided

All cell lines need to be authenticated

Figure 4 : title has a typo mistake "meChanismS"

The title of the article includes MEK inhibition which is not tested in the manscript

Figure 5 : The legend has a mistake in « rappresentation"

Author Response

REVIEWER 1

Comments and Suggestions for Authors

The authors present new evidences for a central role of ErbB3 phosphorylation during the adaptive resistance to BRAF inhibition in melanoma via an autocrine loop mediated by NRG1 secretion. In addition, they present ErbB3 phosphorylation as an early event which can be measured in Circulating Tumor Cells in patient’s blood and be potentially used as a predictive biomarker to targeted therapy.Although the conclusions of the first message are clear and supported by the results, the second message (CTC assay) looks weak and requires additional experiments to confirm the expected effect.

Answer: We thank the reviewer for the positive evaluation of our manuscript. Here below we provide a point-by-point response to the reviewer comments.

Overall, the switch from one cell line to another according to what we are looking at, is disturbing. It would be clearer if all 3 lines are used in all figures.

Answer: We understand the point made by the reviewer. It has to be taken into account that the key experiments of the paper are those represented in Fig 1 and 3, which have been made on all three principal cell lines, namely WM266, WM115 and LOX IMVI. Unfortunatelly, due to the short time allowed for the revision of the manuscript we have been unable to complete an entire set of new experiments. In order to follow at least in part the reviewer comment we have only been able to add a new colony formation assays on WM266 and WM115 cells (please see new Supplementary Figure s4) in support to the finding that BRAFi and anti-NRG1 combinatorial treatments, especially at low doses, potentiate melanoma cell growth inhibition as compared to BRAFi monotherapy.

 Major revisions need to be performed

 MAJOR comments:

Figure 1:

How long were the cells maintained before conditioned medium collection? How long lasted the starvation? L94: the sentence should be more moderate. The words “clearly” and “strong” may be too strong. ErbB3 phosphorylation is only strong in WM115. Fig 1b: It would be appreciated to include a time point 12H for the kinetic of BRAFi on phospho ErbB3. If we relate these results to these in Fig 3: in WM115, the phosphoErbB3 pic of activation is at 8h in Fig 1, but at 12h/24h in Fig 3. Also in LOX IMVI, the pic of activation if at 24h in Fig 1 (there is no 12h to compare) but already at 12h in Fig 3.

Can the authors explain this discrepancy?

The studies were done in the following manner. Cells were seeded and the following day they were serum starved them over night. Afterwards cells were exposed to vemurafenib for the indicated time points (2,8 and 24 hours). Conditioned medium (CM) was collected at each time point and transferred to treatment naive serum starved melanoma cells for 30'. These technical details have been added to the materials and methods section of the revised version of the manuscript (see lines 359-362)                                                    

As to the comment about line 94, we agree with the reviewer criticism and we have corrected the text accordingly (line 96) .                                                                                                                

Regarding the request of the reviewer to add a time point at 12, this would have required repeating the entire experiment and we have decided not to do it in the interest of time. However we would like to point out that the differences noticed by the reviewer in the timing of the peak of activation may derive from the natural variability of the behaviour of each cell and essentially do not change the general message of the experiment.

Figure 2 :

L125: the word “proliferation” is somehow misleading. ATP is measured to estimate cellular metabolic activity (usually correlated to viability/ cell number). It is not a proliferation assay per se. Actually, the results of Fig 2C/D would strengenth a lot by an additional functional assay such as colony formation assay or “real” proliferation assay (EdU/ BrdU or a CFSE cell proliferation assay) Additionnally, these experiments would need to be repeated in at least a second cell line like WM115 where the activation of ERBB3 is stronger (according to Fig 1B). L118 : the kinetik of released NRG1 does not correlate to NRG1 RNA expression of heatmap in Fig 3. Actually, protein release (pic at 8h) appears before RNA (pic at 12h). Fig 2B: is the condition BRAFi + anti NRG1 missing? This condition would show us if BRAFi stimulation for 24h effectively leads to an autocrine loop of NRG1. In addition, wouldn’t we expect a synergistic effect (increase) on phosphoERBB3 under CM+BRAFi compared to BRAFi alone? Fig 2c: the major differences in ATP content are starting at 10nM and lower, it is therefore surprising to also see an effect at 0.35uM. Could the authors explain this fact?

We thank the reviewer for the critisicm regaring our use of the word “proliferation” when using ATP measurement assays. We have corrected the text accordingly (line 128).

To answer the following point, as mentioned above, we have now peformed colony formation assays in WM266 and WM115 cells treated either with BRAFi monotherapy or with the combination of BRAFi and anti-NRG1. Results confirm our previous data regarding the stronger inhibitory effects exerted by the combination. These data have been included in a new figure (supplementary figure s4a and b) of the revised version (see text at lines 130-131)

As to the differences noticed by the reviewer between the peak of NGR1 release vs that of NRG1 RNA expression this is a subtle time differences which may be caused by the variability of the biological system or also from the assays used.

In fig 2B the condition BRAFi + anti NRG1 is indeed missing. However, we have already performed this experiment in Fattore et al (JTM 2013), where we demonstrated that anti-NRG1 was able to abrogate phospho ErbB3 activation following exposure to a BRAFi.

We do not expect a synergistic increase of phospho ErbB3 under CM+BRAFi compared to BRAFi alone because most likely a single condition leads already to plateau receptor phosphorylation.

As to the comment about Figure 2c, we indeed observe statistically significant differences between the two treatments mainly at low doses. We do not have an explanation for the difference observed at the single high dose of 0.35uM. However we believe this does not alter the message of the experiment.

Figure 3 

According to Fig 2, the effect of BRAFi (via NRG 1 release) on ATP content is starting at 10nM and lower, therefore why every other experiments have been done with 0.5uM BRAFi treatment? This should be discussed in detail. Fig 3b: It is not clear what values were used to draw the heatmap. Increase or decrease compared to what ? If the reference was untreated condition, like in the fig S5, the whole first column of the heatmap should have the same color. Please explain better. If we compare RNA and protein expression in Figure 3, they are not always correlated. Example: WM266 untreated has high ERbB3 protein but low RNA. Also, ErbB3 protein decreases until 72h but RNA increases until the pic at 48h (opposite) In WM115, ErbB 3 protein is stable until 72h but RNA increases with pic at 48h LOXIMVI: why is there a protein doublet of ErbB3? This protein increases until a pic at 12h but rna starts medium, decreases early and then increases until pic at 48h The text L152 - 156 does not correctly describe the shown results. Phospho ErbB3 follows similar pattern in all 3 lines with a pic at 12H/24h (except in WM266 where it starts earlier). Suddently, 2 extra cell lines appear (A375 and M14). For which purpose ?

We decided to use a dose of 0.5 uM of BRAFi for western blotting experiments because different cell lines have different degrees of sensitivity to BRAFi and this drug concentration is able to inhibit cell growth efficiently in all cell lines used (see also Fattore et al 2013 JTM and 2015 Oncotarget). We have introduced a sentence to explain this in the text of the revised version at lines L90-93.

Regarding the heat map in Figure 3b, the reviewer is correct. We compared increase/decrease of mRNA expression as shown in supplementary figure s5 as fold change vs untreated condition. Hence, we have now corrected the Figure 3b drawing all the first columns of the heat map with the same colour, i.e. green.

Regarding ErbB3 protein and RNA correlation in Figure 3 we are aware that they are not always correlated in our experiments. We can only speculate that ErbB3 protein/RNA may be regulated at multiple levels, i.e. not only transcitpional but also post-transcriptional. We have now modified the text according to reviewer's suggestion and hope to have better described the findings shown in Figure 3 (lines L156-161).

As to the ErbB3 protein doublet evident in LOX IMVI cells in our western blot analyses we think that it could be related to specific post-transductional modification of the receptor. Interestingly, a similar finding is also present in other publications (see for example Hedge et al, Science Translational Medicine 2013).

The two additional lines tested in this figure, i.e. A375 and M14, were added to provide further evidence of the heterogeneity in the response to BRAFi.

Figure 5 :

Although this CTC kit has been already described, it is based on physical properties of the circulating cells (i.e. size exclusion). Therefore it is highly recommended to confirm the characterization of the filtered cells as CTC (genomic or transcriptomic profiling for example, to check if they are tumor cells at all). How can we exclude the presence of contaminating blood cells ? Could they be counterstained in the immunostaining with a known leucocyte marker for instance? On which basis is the disctintion between pore and actual cell made with the DAPI staining ? Attention, the time of treatment in the patients is confusing. L237 states 72h but in the figure 5a, it is stated 2-3d and in the Fig 5b: 2d. L250 - 255: confusing description of results on erb activation. Are the authors describing the expression of ERBB3 or its activation ? The staining of phosphoERBB3 should always be compared to a total ERBB3 antibody staining to describe the actual activation of the protein. The immunofluorescence of phosphoERBB3 in the zoomed in cell in the treated condition does not show a positive staining. Why do the authors claim otherwise ?

We thank the revisor about the questions regarding CTCs isolation. First of all, ScreenCell kit presents a specific buffer (FC2 buffer) which is able to cause lysis of erythrocyte. This, while reducing the purity of the sample, does not represent a problem for DAPI staining since these cells do not have a nucleus.

Leucocytes can be present in the filter because their size is larger than that of the pores. In order to distinguish these cells from CTCs it is recommended to stain them with an antibody direct against their specific receptor called CD45. In our samples we observed a very low proportion of white cells (see figure s9 panel B). In addition, in the filter areas where we detected both leucocytes and CTCs, only the first showed CD45 positivity as evident in our new supplementary figure s9b as well as in other published work (see Nicolazzo C. et al 2018, Cancers). Conversely, CTCs resulted not positive for CD45. Moreover the DAPI dye used to stain cell nuclei can remain trapped in the pores causing their coloring by autofluorescence (Kuvendjiska et al.2019 Cancers). Despite this issue, the different size of the CTCs and pores is evident enough to allow their distinction. We have better explained these aspects in the results section of the revised version of the manuscript from line L253-261.

Regarding the time of treatment of patients we decided to report exactly when the patients came back to the clinic for the collection of blood samples. Some patients were available after 2 days from initiation of therapy, others after 3 days. This different possibility was included in the clinical protocol.

Regarding the possibility to perform a double staining for pERBB3 and total ERBB3 we have decided not to do it since co-stainings will reduce the interpretation of our results by inducing additional unspecific signals on the filters.

Finally we apologize for the difficulty to evaluate our results for pERBB3 positivity. To help the reader we included additional pictures in the new supplementary figure supplementary s9 panel A.

Figure 6 :

Again, Fig6a is based on the analysis if the previous immunofluorescence staining (Fig 5b) which leads to serious doubts about what cells are we looking at and are they really positive for phosphoErbB3 ? Fig 6c and line 254: 3 out of 11 (30%) patients show a decrease in phosphoErbB3 signal after treatment, what does that mean for the accuracy of the prediction? In the text, the autors talk about « kinase inhibitor » treatment, because 2 different combinations were given to the patients. The 2 combination treatments should be mentioned in the text (or in the methods). According to Table S1, 4 patients had previous treatment with check point inhibitors. Since it is most likely that immunotherapy impacts the patients’ response to additional targeted therapy, the authors should discuss that there might be a biais in the patient cohort’s selection.

We hope we have previously answered to the reviewer’s comments and clarified how we have counted CTCs characterized by pERBB3 positivity. As rightly noted by reviewer we observed that 3 patients out of 11 show decreased receptor phosphorylation. We do not have an explanation about this albeit we have to take into account that melanoma is a highly heterogeneous disease. Obviously, this was a pilot study which will need to be repeated in a large cohort of patients. Nevertheless we find very promising to have found ErbB3 phosphorylation increase after treatment in a statistically significant manner.

In agreement with the reviewer, we modified «kinase inhibitor» with « BRAF and MEK inhibitors  or BRAF/MEK inhibitors» in the text as reported in red one color. 

Finally, as rightly noted by the reviewer, we are aware that some melanoma patients had been previously treated with check point inhibitors (i.e. 4 of 11). Again, this was a pilot study and was not designed to allow to detect differences between patients previosly treated with immune checkpoint inhibitors vs untreated.  This aspect is interesting to be included in a future study on a larger cohort of patients.

Discussion

ErbB2/ErbB3 heterodimer binds to other ligands than NRG1. Should it be discussed which ones and their potential role in the BRAFi resistance ? 

The reviewer is correct. The ErbB3 receptor is known to be activated by several members of the Neuregulin family. These proteins derive from the alternative splicing of four genes giving rise to at least 26 different isoforms (Mota et al, Oncotarget, 2017 Previous publications (see for Capparelli et al, JBC 2015 and Montero-Conde et al, Cancer Discovery 2013) pointed out to NRG1 as the major ligand involved in drug resistance as we do in our manuscript (lines 297-301). To the best of our knowledge, no study has investigated the role of the other ligands of NRG family in this context and this could be an aspect to be further investigated in the future. It has to be pointed out however that our neutralization experiments with antibodies against NRG1 further underscore the importance of this particular isoform.

L287 - 289: if this finding is the most important as the author claim, it should probably be backed-up by another functional assay in addition to ATP measurement (as already mentionned)

The reviewer is right. We strongly hope that our new clonogenic data have strenghtened this finding.

L289-292 : the central role of ErbB3 in this autocrine loop could be tested by gene silencing under the CM + anti-NRG 1 condition.

We thank the reviewer for the suggestion. However we have already demonstrated the role of the ErbB3 receptor in the past through the use of neutralizing monoclonal antibodies (Fattore et al, Oncotarget 2015)

L327-328: It is still difficult to claim this, based on the data of figure 5 and 6.

We strongly hope to have clarified in our previous point by point answers the reasons which brought us to those conclusions.

Suggestion to enhance the clinical relevance of the data :

Could the author verify a potential correlation between erbB3 activity and melanoma patient survival (if not already done) ?

The melanoma patient cohort of the TCGA provides survival data on 470 patients (https://portal.gdc.cancer.gov/) and proteomic data (phosphoErbB3) from TCGA seem to be available according to this reference ( Li J, Lu Y, Akbani R, Ju Z, Roebuck PL, Liu W, et al. TCPA: a resource for cancer functional proteomics data. Nat Methods. 2013;10:1046–7.)

As an alternative, a study on melanoma patients with BRAFi+MEKi and RNAseq data available (such as GSE50509 or more recent) would provide the expression status of a panel of Erbb3 transcriptional target genes (identified here : Mahmoud El Maassarani et al ; PLOS one 2016)

We thank the reviewer for the suggestions. ErbB3 expression levels have been already demonstrated to be correlated with worst survival in melanoma (Tiwary S. et al. Oncogenesis. 2014 ; Reschke M. et al. Clinical Cancer Research, 2008 ). Regarding the correlation between ErbB3 activity and melanoma patients survival there are no evidences in the literature. We thank the reviewer to have suggested us to check this aspect in TCGA data available at the reference (Li et al, Nat Methods 2013). We did it and we obtained the Kaplan-Meyer curve below which shows that ErbB3 activity does not affect survival probability.

Minor comments:

Supplementals

Fig s2: statistical analysis should be provided

We have added results of statistical analysis in the new version of supplementary figure s2

Fig s4: wm266: attention, the blot of gapdh does not correspond to the one in Fig 3a. Which one is the correct loading control? this is quite important because if the correct loading control is that on Fig S4, then it is unclear if ErbB3 is phosphorylated at all under BRAFi treatment.

Reviewer is right. We have now corrected our previous mistake replacing the correct GAPDH in the new Figure 3a and in the new supplementary Figure s5 which corresponds to previous supplementary figure s4.

Fig s5: statistical analysis should be provided

We have added results of statistical analysis in the new Figure s6 which corresponds to previous supplementary figure  s5

All cell lines need to be authenticated

We provided details about their authentication in a recent publication by our laboratory (Fattore et al, 2018 CDD)

Figure 4 : title has a typo mistake "meChanismS"

We have corrected this mistake

The title of the article includes MEK inhibition which is not tested in the manscript

We have decided to modify the tile and replace the wording “BRAF/MEK inhibitors” with “targeted therapy”. At any rate, it has to be taken into account that in our previous work (see Fattore et al, 2013 JTM and Fattore et al, 2015 Oncotarget) we showed that the activation of ErbB3 is an early adaptive event taking place in melanoma cells both in response to BRAF and MEK inhibition. Furthermore, some patients from whom CTCs were analyzed had been treated with BRAF and MEK inhibotors combo therapy

Figure 5 : The legend has a mistake in « rappresentation"

We have corrected this mistake

Reviewer 2 Report

This paper focuses on the role of the ErbB3 receptor in BRAF-mutated melanoma cell lines treated with a BRAF inhibitor (BRAFi). They showed that a BRAFi or the conditioned medium coming from BRAFi-stimulated melanoma cells induced ErbB3 and AKT phosphorylation due to release of NRG-1 by the melanoma cells. NRG-1 inhibition potentiated the growth inhibitory effect of BRAF inhibition. Five BRAF-mutated melanoma cells show different kinetics of ErbB3 and NRG-1 activation after treatment with a BRAFi. NRG-1 activation occurs via different mechanisms in two related cell lines. Finally, pErbB3 upregulation could be observed in circulating melanoma cells after treatment of melanoma patients with BRAF+MEK inhibitors.

This work is interesting for the understanding the role of NRG-1/Erbb3 in resistance to targeted therapies. However, some points need to be addressed.

Although the title indicates “resistance to BRAF/MEK inhibitors “ no MEK inhibitors were used in any of the ex vivo experiments making it difficult to correlate the ex vivo data with the CTC data coming from patients treated with BRAF and / or MEK inhibitors.

Figure 2C: A very low dose of vemurafenib (10nM) is necessary to see a significant effect of the anti-NRG1 peptide. Is ERK inhibition still efficient at this low dose?

Figure 4: Could the authors explain the discrepancy between Fig4A-WM266 (no luciferase induction with BRAFi) and Fig4B-WM266 (increase in NRG1 mRNA with BRAFi). The choice of WM115 and WM266 cell lines for these experiments is striking as both cell lines were derived from the same patient: WM115 cell line originated from the primary tumor, WM266-4 were from a lymph-node metastasis. Could this explain the difference in results between both cell lines? Can a similar result be obtained with another metastatic cell line?

Figure 5: Explain in more details the CTC figures. What are the non-CTC cells showing a very intense red signal? Is this non-specific? Why is the pERBB3 labelling so weak on CTC?

Author Response

REVIEWER 2

Comments and Suggestions for Authors

This paper focuses on the role of the ErbB3 receptor in BRAF-mutated melanoma cell lines treated with a BRAF inhibitor (BRAFi). They showed that a BRAFi or the conditioned medium coming from BRAFi-stimulated melanoma cells induced ErbB3 and AKT phosphorylation due to release of NRG-1 by the melanoma cells. NRG-1 inhibition potentiated the growth inhibitory effect of BRAF inhibition. Five BRAF-mutated melanoma cells show different kinetics of ErbB3 and NRG-1 activation after treatment with a BRAFi. NRG-1 activation occurs via different mechanisms in two related cell lines. Finally, pErbB3 upregulation could be observed in circulating melanoma cells after treatment of melanoma patients with BRAF+MEK inhibitors.

This work is interesting for the understanding the role of NRG-1/Erbb3 in resistance to targeted therapies. However, some points need to be addressed. 

We thank the reviewer for the positive evaluation of our work. Here below we have provided a point-by-point response to the reviewer specific comments.

Although the title indicates “resistance to BRAF/MEK inhibitors “ no MEK inhibitors were used in any of the ex vivo experiments making it difficult to correlate the ex vivo data with the CTC data coming from patients treated with BRAF and / or MEK inhibitors. 

We have decided to modify the tile and replace the wording “BRAF/MEK inhibitors” with “targeted therapy”. At any rate, it has to be taken into account that in our previous work (see Fattore et al, 2013 JTM and Fattore et al, 2015 Oncotarget) we showed that the activation of ErbB3 is an early adaptive event taking place in melanoma cells both in response to BRAF and MEK inhibition. Furthermore, some patients from whom CTCs were analyzed had been treated with BRAF and MEK inhibitors combo therapy

Figure 2C: A very low dose of vemurafenib (10nM) is necessary to see a significant effect of the anti-NRG1 peptide. Is ERK inhibition still efficient at this low dose?

We have previously shown (see Fattore et al 2013 JTM) that vemurafenib is able to inhibit pERK signaling and in turn to induce pErbB3/AKT signaling in a dose dependent way also at low nanomolar doses.

Figure 4: Could the authors explain the discrepancy between Fig4A-WM266 (no luciferase induction with BRAFi) and Fig4B-WM266 (increase in NRG1 mRNA with BRAFi). The choice of WM115 and WM266 cell lines for these experiments is striking as both cell lines were derived from the same patient: WM115 cell line originated from the primary tumor, WM266-4 were from a lymph-node metastasis. Could this explain the difference in results between both cell lines? Can a similar result be obtained with another metastatic cell line?

We understand the reviewer’s concern regarding the discrepancy in the behaviour of WM115 and WM226, in the light of the fact that these cells derive from the same patient. We don’t have a definitive explanation for this finding. As the reviewer suggests one possible explanation may be the different origin, i.e. primary vs metastatic of the two cell lines. Inspired by the reviewer comment we have performed the same luciferase experiments also in another metastatic cell line, i.e. A375.  Interestringly we do not observe a significant increase in the activity of NRG1 promoter following BRAF inhibition as in WM266 cells (see new supplementary Figure s7). We have included a comment about this finding in the revised version of the manuscript (L207-209).

Figure 5: Explain in more details the CTC figures. What are the non-CTC cells showing a very intense red signal? Is this non-specific? Why is the pERBB3 labelling so weak on CTC?

The intense brigth spots in the figure are not CTC cells but rather non-specific signals. The reason is that some amount of the fluorescent anti-pErbB3 antibody remains trapped in the pores causing their coloring by autofluorescence and this event can cause the appearance of non-specific signals in the filter (see also Kuvendjiska et al.2019 Cancers). We have added this explanation in the text of the revised version of the manuscript (lines 253-261). Regarding the weak signal of pErbB3 on CTCs we agree with the reviewer about the overal weekness of the specific signal. To further support our data we have inserted additional pictures in the new supplementary figure s9 panel A. Zoomed pERBB3 positive CTCs are shown as additional examples.

Round 2

Reviewer 1 Report

FigS4:

quantification of CFA can not be called "% proliferation" but "% cell growth"

Fig 3:

ErbB3 protein doublet : what is a post-transductional modification exactly ? Do the authors mean post-translational ?

Author Response

Comments and Suggestions for Authors

FigS4:

quantification of CFA can not be called "% proliferation" but "% cell growth"

We thank the reviewer for the suggestion. We have now corrected "% proliferation" with "% cell growth" in FigS4.

Fig 3:

ErbB3 protein doublet : what is a post-transductional modification exactly ? Do the authors mean post-translational ?

The reviewer is right. In the previous document concerning Figure 3 there was a typing mistake. As the reviewer rightly noted, we meant to write "post-translational"
